# Discovery of the Inhibitor Targeting the SLC7A11/xCT Axis through In Silico and In Vitro Experiments

**DOI:** 10.3390/ijms25158284

**Published:** 2024-07-29

**Authors:** Jianda Yue, Yekui Yin, Xujun Feng, Jiawei Xu, Yaqi Li, Tingting Li, Songping Liang, Xiao He, Zhonghua Liu, Ying Wang

**Affiliations:** 1The National and Local Joint Engineering Laboratory of Animal Peptide Drug Development, College of Life Sciences, Hunan Normal University, Changsha 410081, China; 201630151080@hunnu.edu.cn (J.Y.); ykyin2021@hunnu.edu.cn (Y.Y.); 201630012032@hunnu.edu.cn (X.F.); 202070141023@hunnu.edu.cn (J.X.); yqli@hunnu.edu.cn (Y.L.); 17537202881@163.com (T.L.); liangsp@public.cs.hn.cn (S.L.); 2Peptide and Small Molecule Drug R&D Plateform, Furong Laboratory, Hunan Normal University, Changsha 410081, China; 3Institute of Interdisciplinary Studies, Hunan Normal University, Changsha 410081, China; 4Shanghai Engineering Research Center of Molecular Therapeutics and New Drug Development, School of Chemistry and Molecular Engineering, East China Normal University, Shanghai 200062, China; xiaohe@phy.ecnu.edu.cn; 5Shanghai Frontiers Science Center of Molecule Intelligent Syntheses, School of Chemistry and Molecular Engineering, East China Normal University, Shanghai 200062, China; 6New York University–East China Normal University Center for Computational Chemistry, New York University Shanghai, Shanghai 200062, China

**Keywords:** SLC7A11/xCT axis, virtual screening and molecular dynamics simulation, cervical cancer, oxidative stress, drug discovery

## Abstract

In the development and progression of cervical cancer, oxidative stress plays an important role within the cells. Among them, Solute Carrier Family 7 Member 11 (SLC7A11/xCT) is crucial for maintaining the synthesis of glutathione and the antioxidant system in cervical cancer cells. In various tumor cells, studies have shown that SLC7A11 inhibits ferroptosis, a form of cell death, by mediating cystine uptake and maintaining glutathione synthesis. Additionally, SLC7A11 is also involved in promoting tumor metastasis and immune evasion. Therefore, inhibiting the SLC7A11/xCT axis has become a potential therapeutic strategy for cervical cancer. In this study, through structure-based high-throughput virtual screening, a compound targeting the SLC7A11/xCT axis named compound 1 (PubChem CID: 3492258) was discovered. In vitro experiments using HeLa cervical cancer cells as the experimental cell model showed that compound 1 could reduce intracellular glutathione levels, increase glutamate and reactive oxygen species (ROS) levels, disrupt the oxidative balance within HeLa cells, and induce cell death. Furthermore, molecular dynamics simulation results showed that compound 1 has a stronger binding affinity with SLC7A11 compared to the positive control erastin. Overall, all the results mentioned above indicate the potential of compound 1 in targeting the SLC7A11/xCT axis and treating cervical cancer both in vitro and in silico.

## 1. Introduction

Cervical cancer is one of the most common gynecological malignancies, with high incidence and mortality rates, posing a serious threat to women’s health and life [1,2,3,4]. Elucidating the mechanisms underlying cervical cancer development and developing effective therapeutic strategies are of great significance for the prevention and treatment of cervical cancer [5,6]. The occurrence and progression of cervical cancer are closely related to changes in cellular homeostasis, with oxidative stress playing a key role in this process [7,8,9]. To adapt to the highly oxidative environment, tumor cells need to enhance their antioxidant system to maintain cellular redox balance [10,11,12]. Solute carrier family 7 member 11 (SLC7A11), also known as xCT, is a cystine/glutamate transporter that mediates the uptake of exogenous cystine by tumor cells [13,14,15]. Under the action of SLC7A11, exogenous cysteine enters the cell and undergoes a series of reactions. Initially, it is transformed into cystine and subsequently reacts with substances such as glutamate to generate the direct precursor of glutathione, γ-glutamylcysteine. Ultimately, through reactions with substances including glycine, glutathione (GSH) is formed. GSH interacts with intracellular ROS to produce glutathione disulfide (GSSG), thereby maintaining intracellular redox balance and reducing the impact of reactive oxygen species on cells. This mechanism is crucial for the survival of cancer cells [14,15,16,17] (the specific mechanism of SLC7A11/xCT in maintaining intracellular glutathione synthesis is shown in Figure 1).

SLC7A11 is significantly upregulated in various types of cancers [18]. Research has found that SLC7A11 can inhibit ferroptosis of cancer cells [13,15,17,18], including cervical cancer cells [16,19,20], thereby promoting tumor development. Specifically, ferroptosis, as a distinct form of cell death, is mechanistically associated with the accumulation of intracellular iron ions and lipid peroxidation. In ferroptosis, the excessive accumulation of iron ions within cells leads to lipid peroxidation, resulting in the disruption of cell membrane integrity and eventual cell demise. Therefore, the maintenance of intracellular redox balance is crucial for mitigating ferroptosis. Importantly, SLC7A11 plays a pivotal role in this process by facilitating the synthesis of glutathione, which serves as a key antioxidant within cells. By promoting glutathione synthesis, SLC7A11 contributes to the preservation of intracellular redox homeostasis, thereby attenuating the occurrence of ferroptosis [13,14,15,21]. In addition, SLC7A11 can promote tumor metastasis [22]. SLC7A11 also contributes to glutathione-based drug resistance [21,23,24,25].

In summary, inhibiting the SLC7A11/xCT axis may be a potential strategy for the treatment of cervical cancer. Currently, some targeted inhibitors have been developed, such as erastin [26,27,28,29] and HG106 [21]. These inhibitors can interfere with the function of SLC7A11, inhibiting the proliferation and evasion of apoptosis of cervical cancer and many other cancer cells [21,26,27,28,29]. However, these inhibitors have certain limitations and shortcomings, including high effective concentrations, limited in-depth research, and poor selectivity. Moreover, there are very few targeted inhibitors of the SLC7A11/xCT axis, and no compounds have entered clinical research. Therefore, further research and development of new targeted inhibitors are needed to improve the effectiveness and clinical prospects of anti-cervical cancer research.

This study aims to discover targeted inhibitors of the SLC7A11/xCT axis, through 77 structure-based high-throughput virtual screenings, that would potentially be able to disrupt the redox balance in cervical cancer cells and inducing cell death. Based on the aforementioned objectives, this study uses the HeLa cervical cancer cell line as a cellular experimental model to explore the effects of candidate compounds obtained from virtual screening on intracellular glutathione, glutamate, and ROS levels in HeLa cells, as well as determine the toxicity and anti-migration activity of small molecules with potential to inhibit the SLC7A11/xCT axis. Furthermore, molecular dynamics simulation was used to analyze and compare the affinity of compounds with the positive control erastin for SLC7A11. In summary, this study aimed to discover and validate potential SLC7A11/xCT axis inhibitors computationally and experimentally.

## 2. Results

In this work, compound 1, a targeted inhibitor of the SLC7A11/xCT axis, was discovered through structure-based high-throughput virtual screening of the Specs database (https://www.specs.net/, accessed on 1 May 2022). Further experiments showed that compound 1 has the potential activity to reduce glutathione levels in HeLa cells, thereby disrupting the redox balance in cervical cancer cells and inducing cancer cell death at the two- and three-dimensional levels. Compound 1 can also inhibit HeLa cell migration. Furthermore, molecular dynamics simulations demonstrated the stable binding of the compound to SLC7A11. Here we list the following results and discussion.

### 2.1. The Discovery of Potential SLC7A11/xCT Axis Inhibitors

As illustrated in Figure 2, this study employed molecular docking-based virtual screening to identify potential inhibitors targeting the SLC7A11/xCT axis from the Specs database, which contains over 200,000 compounds. Initially, all compounds from the Specs database were docked individually with SLC7A11 (PDB ID: 7EPZ, resolution 2.01 Å) using Glide Standard Precision (SP) and Extra Precision (XP). The top 5000 compounds based on docking scores were selected from each docking result.

Subsequently, the obtained two sets of compounds were filtered to retain only those fully compliant with Lipinski’s rule [30,31,32] of five (molecular weight < 500, logP < 5, hydrogen bond donors < 5, hydrogen bond acceptors < 10) and Jorgensen’s rule [33,34] (hydrogen bond acceptors > 1, rotatable bonds < 8). Eventually, 989 (SP docking) and 1410 (XP docking) compounds meeting the criteria were obtained. The intersection of these two sets yielded 144 common candidate compounds.

Next, the water solubilities of these 144 compounds were evaluated using SwissADME [35] (http://www.swissadme.ch/, accessed on July 2022), and compounds with poor solubility were discarded, resulting in 83 small molecules with good solubility. Subsequently, the interaction energies (Eint) between these 83 compounds and the key residues of SLC7A11 were analyzed (derived from Schrodinger software (version 2022-1). It was observed that regardless of the docking method, the mean Eint between the compounds and the ARG 340 residue was the lowest, indicating the strongest interaction with this residue. This suggests that ARG 340 may be a key residue determining ligand affinity and selectivity.

Finally, using the median Eint between the compounds and the ARG 340 residue as the threshold for both docking methods (SP and XP), 20 compounds (see Appendix A) with Eint values below this threshold were selected. These compounds represent potential candidates for high-affinity inhibitors of SLC7A11 and will be further validated and optimized in subsequent experimental studies.

### 2.2. In Vitro Evaluation of the Effect Compound 1 Has on the Oxidative/Antioxidant Balance within HeLa Cells or Similar

The compounds obtained through high-throughput virtual screening were purchased from TOPSCIENCE (USA) with 19 out of the 20 compounds successfully acquired. Compound 20 was not obtained due to lack of availability. The molecular compositions of the 19 compounds were detected using H-NMR or LC-MS, and the specific molecular compositions of them are shown in Appendix A. Meanwhile, the 2D structures of compound 1 and erastin are also shown in Figure 3I,J, respectively. Before further investigation, the effects of these 19 compounds on intracellular glutathione (GSH) levels in HeLa cells were tested to identify effective small molecules targeting the SLC7A11/xCT axis. All compounds were administered at a concentration of 15 μM. Compared to the control group, only compound 1 was able to decrease intracellular GSH levels by approximately 22%, while the remaining compounds had no effect on intracellular GSH levels (see Appendix A). Therefore, compound 1 was selected for further investigation and used alongside the SLC7A11 inhibitor erastin as a positive control.

Given that compound 1, at 15 μM, only reduced intracellular GSH levels in HeLa cells by approximately 22%, the dosage of compound 1 and erastin was increased to 25 μM for further testing. After 24 h of drug incubation, both 25 μM compound 1 and erastin significantly reduced intracellular GSH levels in HeLa cells (see Figure 3A), with compound 1 also increased intracellular glutamate levels (see Figure 3B). Furthermore, flow cytometry analysis of reactive oxygen species (ROS) revealed that both 25 μM compound 1 and erastin significantly increased intracellular ROS levels in HeLa cells (see Figure 3F). These results suggest that compound 1, at a certain concentration, promotes oxidative stress in HeLa cells by targeting the SLC7A11/xCT axis.

As shown in Figure 3C, cytotoxicity assays indicated that compound 1 exhibited delayed cytotoxicity, as even at a concentration of 100 μM, its viability-reducing effect on HeLa cells after 24 h of incubation did not reach 50%. However, after 36 h of incubation, compound 1 demonstrated a substantial IC_50_ of approximately 10.23 μM, indicating its toxicity against HeLa cells. Additionally, longer drug exposure (48 h) did not significantly enhance compound 1’s toxicity towards HeLa cells (with a corresponding IC_50_ of approximately 10.51 μM). Thus, treatment with compound 1 for 36 h proved to be the optimal duration for exerting its toxic effects. Furthermore, at 36 h of incubation, compound 1 exhibited comparable cytotoxicity to erastin at 50 μM, but at lower concentrations (25, 12.5, and 6.25 μM), compound 1 suppressed HeLa cell viability to below 40%, with viability dropping below 60% at 3.13 μM. In contrast, at equivalent concentrations, erastin maintained HeLa cell viability at approximately 50%, 70%, 80%, and 80%, respectively (see Figure 3E).

Inhibiting the SLC7A11/xCT axis disrupts cystine uptake by cells, leading to decreased intracellular cystine levels, reduced availability of glutathione biosynthetic precursors, and ultimately cell death due to excessive oxidative stress [13,14,15,21] (see Figure 1). As depicted in Figure 3D, supplementation with 2.5 mM L-cystine during compound 1 incubation at various concentrations (12.5, 6.25, and 3.13 μM) significantly restored HeLa cell viability, indicating that L-cystine compensated for intracellular cystine levels and reaffirming compound 1’s targeting of the SLC7A11/xCT axis and disruption of the oxidative/antioxidant balance in HeLa cells.

Lastly, when HeLa cells were co-treated with compound 1 at various concentrations and 5 or 10 mM of the ROS scavenger N-acetylcysteine (NAC) for a certain concentration gradient of compound 1, the combined treatment significantly increased HeLa cell viability compared to treatment with compound 1 alone (see Figure 3G), further supporting the role of ROS in mediating the cytotoxic effects of compound 1. Additionally, visual assessment of calcein staining results provided further evidence for the aforementioned findings (see Figure 3H).

Taken together, these results demonstrate that compound 1 disrupts the oxidative/antioxidant balance in HeLa cells by targeting the SLC7A11/xCT axis, leading to a fatal increase in ROS levels and subsequent cell death.

### 2.3. In Vitro Evaluation of the Inhibitory Effects of Compound 1 on HeLa Cell Migration at Low Concentrations

Inhibiting tumor migration is of significant importance in the development of anti-cancer drugs. In this study, erastin at concentrations of 0.5, 1, 2, and 4 μM, along with compound 1, was selected to analyze the inhibitory effects of compound 1 on HeLa cell migration. According to the results of the scratch wound healing assay (Figure 4A,B), after 24 h of drug treatment, compared to the control group, all concentrations of erastin did not show a significant trend in inhibiting the healing of HeLa cells. However, compound 1 at concentrations of 1, 2, and 4 μM significantly reduced the healing rate of HeLa cells, by approximately 10%, 13%, and 14%, respectively. Additionally, compound 1 at a concentration of 0.5 μM did not significantly reduce the healing rate of HeLa cells. In the Transwell migration assay (Figure 4C,D), compared to the control group, both 4 μM compound 1 and erastin significantly inhibited the migration activity of HeLa cells, with corresponding migration rates reduced by approximately 94% and 99%, respectively. In summary, compound 1 effectively inhibits the migration of HeLa cells at low concentrations.

### 2.4. Assessment of the Toxic Effects of Compound 1 on HeLa Three-Dimensional Spheroids

To further demonstrate the cytotoxic effect of compound 1 on HeLa cells, this study conducted three-dimensional (3D) tumor spheroid experiments, as spheroid models better reflect the complex tumor microenvironment in vivo compared to two-dimensional cell models. HeLa spheroids were treated with 10 μM erastin or 5 μM and 10 μM compound 1 in a 96-well round-bottom ultra-low-attachment plate, with the treatment starting on day 0. Photographs and measurements of HeLa spheroid volume were taken daily from day 0, recording a total of six consecutive measurements of tumor spheroid volume (Figure 5A). On the second day of treatment (Day 1), there was a noticeable difference in spheroid volume between the three treatment groups and the control group (Figure 5A,B), indicating that both erastin and compound 1 significantly reduced the proliferation rate of HeLa tumor spheroids after 24 h of treatment (Figure 5C). Furthermore, it was found that as the duration of drug treatment increased to 48 h (Day 2), the spheroids treated with 5 μM and 10 μM compound 1 were significantly smaller than those in the control and erastin groups (Figure 5A,B), with the proliferation inhibition of 5 μM and 10 μM compound 1 being comparable to that of 10 μM erastin at this point (Figure 5C). However, when the drug incubation time reached 72 h, 5 μM and 10 μM compound 1 were more effective at significantly reducing the proliferation rate of HeLa tumor spheroids compared to 10 μM erastin (Figure 5C). Meanwhile, the volume difference between the HeLa tumor spheroids in all treatment groups and the control group increased over time (Figure 5A,B). Additionally, it is worth noting that 5 μM and 10 μM compound 1 caused irregular growth and unclear edges of HeLa tumor spheroids, whereas 10 μM erastin did not achieve this effect (Figure 5A). In summary, the three-dimensional tumor spheroid experiment further validates the potential application of compound 1 as an inhibitor of SLC7A11, improving the complex tumor environment in vivo.

### 2.5. Molecular Dynamics Simulations

#### 2.5.1. Structural Mobility and Compactness of SLC7A11 Systems

The docking complexes of compound 1 with SLC7A11 (SLC7A11-c1), as well as erastin with SLC7A11 (SLC7A11-erastin), were subjected to continuous 500 ns molecular dynamics simulations, alongside the apo state of SLC7A11 (APO). Prior to the dynamics simulations, the systems were embedded in a cell membrane (see Appendix A).

The CA atoms’ average RMSD values aligned with the initial docked structure were 2.5, 1.6, and 1.8 Å for the APO, SLC7A11-c1, and SLC7A11-erastin complexes, respectively. These systems demonstrated equilibration within the 500 ns simulations, reaching a local potential energy minimum, with RMSD fluctuations remaining below 2.0 Å, as depicted in Figure 6A. Notably, compound 1 exhibited a marginally lower RMSD value compared to erastin, as shown in Figure 6B. Compound 1 demonstrated a relatively stable conformation, with transient fluctuations of approximately 30 ns occurring after 360 ns, before reverting to its initial state and maintaining this conformation throughout the remainder of the simulation. In contrast, the positive control erastin exhibited more pronounced RMSD fluctuations during the equilibration phase.

The analysis of the radius of gyration (Rg) revealed the compactness of the three systems. As shown in Figure 6C, in comparison to APO, all ligands bound to SLC7A11 exhibited a tendency to fluctuate around their initial conformations during the simulation. Specifically, SLC7A11-erastin showed a slight decrease at 50 ns, followed by a gradual increase, stabilizing around 100 ns and maintaining stability until the simulation’s conclusion. SLC7A11-c1 displayed a slow initial decrease, followed by a gradual increase after 100 ns, stabilizing around 230 ns and remaining in this state until the end of the simulation. APO, on the other hand, appeared to progressively compact over time, with a gradual decrease at 330 ns, stabilizing at 370 ns, and maintaining stability for the rest of the simulation. The average Rg values for APO, SLC7A11-c1, and SLC7A11-erastin were 22.8, 22.7, and 22.6 Å, respectively.

The structural flexibility and movements were investigated using alpha carbon atoms’ root mean square fluctuation (RMSF). As depicted in Figure 6E, SLC7A11 encompasses 12 transmembrane (TM) segments (TM1–12), and the residues of each segment are also shown. RMSF analyses revealed major peaks which generally corresponded to TM6 and TM5, as seen in Figure 6D. The RMSF analysis of the APO system revealed a peak at the TM6 region, with the maximum RMSD reaching 3.5 Å. In the case of SLC7A11 bound to the ligand erastin, the RMSF peak was observed at the TM10 region, where the maximum RMSD was recorded at 2.9 Å. For SLC7A11-c1, the RMSF peaks were noted near the TM12 region and the C-terminus, with the maximum RMSD value being 3.5 Å. The RMSF in TM6 and TM5 was significantly higher for the APO compared to the ligand-bound structures, highlighting the importance of these two regions for navigating roles of substrate entry. This finding stands in stark contrast to another computational study [36], which demonstrated that, in a 200 ns simulation, the RMSF values of TM8 were significantly higher when bound to a substrate compared to the APO form. The discrepancy in the results can be attributed to differences in the starting structures (active versus inactive forms). However, the conformational behavior of the helices forming the ligand-binding domain of SLC7A11 warrants further investigation.

#### 2.5.2. Collective Motions of SLC7A11 Systems

PCA can be employed to examine the relationship between different conformations sampled during the trajectory. As shown in Figure 7A, compared to the APO state, the ligand-bound SLC7A11 systems exhibit more pronounced conformational clustering. The SLC7A11 complexed with compound 1 forms clusters along PC1 at +10 and −15, while the SLC7A11 complexed with erastin forms clusters at +10 and −10. Appendix A illustrates the 20 eigenvectors used to calculate the significant movements observed in the protein structure during the simulation. The first three principal components account for 45% and 45.1% of the variance in the SLC7A11 complexes bound to compound 1 and erastin, respectively. In contrast, the first four principal components of the APO state explain 55.5% of the variance. This result further indicates that the APO state exhibits more pronounced motion compared to the ligand-bound systems. Ligand binding influences the motion of residues around the pocket, as evidenced by the heatmap of the first principal component (Figure 7B). Atomic displacements are represented by a color scale from blue to red, indicating low to high displacement, with the size of the arrows representing the extent of motion and the direction indicated by the arrows. Compared to the ligand-bound systems, the residues in the active pocket of the APO system show inward motion and significant atomic displacement. The analysis of residue contributions to the first principal component (Figure 7C) corroborates the findings from RMSF analysis, highlighting substantial fluctuations in the TM6, TM5, and TM12 regions.

#### 2.5.3. Local Correlation Motion Patterns

The degree to which the atomic fluctuations or displacements within a system are correlated with each other can be evaluated by analyzing the magnitude of all pairwise cross-correlation coefficients. Incorporating the insights from RMSF analysis and the residue contribution analysis of PC1 in PCA, it was observed that the residues at TM5, TM6, TM11, and TM12 exhibited the most significant fluctuations. Consequently, a residue range of 184–214, 230–260, and 425–465, totaling 100 residues, was selected to analyze the degree of inter-correlation of atomic displacements within these regions. A two-dimensional representation of local DCCM is depicted in Figure 8, illustrating the correlation between two residues. Correlated motions are indicated by shades of cyan (+0.8 to +1.0), while anti-correlated motions are denoted by shades of pink (−0.8 to −1.0). As depicted in Figure 8A, the APO system illustrates strong positive and negative correlated motions at the protein level. In contrast, the SLC7A11-erastin system exhibits a slightly reduced pattern of correlation and anti-correlation motions compared to the APO system (Figure 8B). The SLC7A11-erastin system (Figure 8C) demonstrates the weakest correlations and anti-correlations. Figure 8D,E represent the correlated motions with line segments, where blue lines denote anti-correlated motions and red lines indicate positively correlated motions. The overall degree of anti-correlation across the three systems is APO > SLC7A11-erastin > SLC7A11-c1. The analysis suggests that the diminished anti-correlation observed in the ligand-bound systems may indicate a closed state of the protein transport channel, potentially inhibiting substrate translocation.

#### 2.5.4. Hydrogen Bond Analysis

In the simulations, the formation and dissolution of hydrogen bonds between SLC7A11 and the ligands were observed and analyzed. A stable hydrogen bond was noted to form between erastin and GLN 191, persisting for the longest continuous duration of 2.3 ns within the simulation time (Figure 9A). Compound 1 was observed to maintain stable hydrogen bond interactions with THR 56 and GLN 191, with the longest continuous time of 2.3 ns and 0.7 ns, respectively, as shown in Figure 9B. The analysis of hydrogen bond occupancy, which represents the temporal fraction of residue participation in hydrogen bond formation within the protein–ligand complex system, revealed key residues. Residues GLN 191, THR 56, LYS 198, and THR 195 were found to significantly contribute to the stability of ligand binding within the SLC7A11-c1 system, as illustrated in Figure 9C. In the SLC7A11-erastin system, the formation of hydrogen bonds with GLN 191 was deemed critical, underscoring the importance of GLN 191 across various ligand-bound states.

#### 2.5.5. K-Means Clustering Analysis

The K-means clustering algorithm was used to detect the conformational changes produced by the complexes after running molecular dynamics simulations, extracting representative dominant structures. Through iterative exploration of various cluster sizes for the two ligand binding systems, a cluster size of 2 was found to be optimal, as evidenced by the lowest Davies–Bouldin index (DBI) value and the highest silhouette score (pSF) (Appendix A), which corroborated the PCA findings. Representative structures were identified from the SLC7A11-c1 and SLC7A11-erastin simulation trajectories. As depicted in Figure 10, the chlorophenoxy group at one terminus of the erastin molecule is projected into a hydrophobic pocket formed by the alignment of TM1A, TM6B, and TM7. At the other extremity of the erastin molecule, the quinazoline moiety resides within a hydrophobic pocket encompassed by TM5 and TM8, engaging in interactions with GLN 191 and PHE 336. It was observed that one end of compound 1 is positioned within the hydrophobic pocket encircled by TM5 and TM8, and further occupies a subsidiary pocket encompassed by TM6A and TM8. Additionally, hydrogen bonds are formed with GLN 191, THR 56, LYS 198, and THR 195. These interactions underscore the importance of specific residues in stabilizing the protein–ligand complexes.

#### 2.5.6. Free Energy Calculations

The binding affinity between inhibitors and their target proteins is governed by the second law of thermodynamics. MM/GBSA was utilized to predict their binding free energy due to its enhanced computational efficiency and practicality in drug screening and design compared to MM/PBSA. This approach dissects the total binding free energy into distinct components, including electrostatic interactions (EEL), van der Waals forces (VDWAALS), and solute–solvent interactions (DELTA G solv). The calculations were concentrated on the final 100 ns of stable trajectories, As depicted in Figure 11A, the average binding energies were found to be significantly similar (with ΔGbind for compound 1 being −40.60 kcal/mol and for erastin being −42.30 kcal/mol). The most favorable interactions were primarily attributed to van der Waals forces, while the least favorable contributions stemmed from solvation terms. Subsequently, a thorough examination of the energy contributions of residues in each ligand binding system was conducted, as shown in Figure 11B. Notably, the energy contribution peaks for residues in SLC7A11-c1 and SLC7A11-erastin were strikingly similar, clustering within the extensive ligand binding domain at TM1, TM5, TM6, and TM8. The residues with significant contributions to the ligand binding energy are depicted in Figure 11C,D. For compound 1, the most favorable residues included THR 56, GLN 191, LYS 198, and LEU 252, while for erastin, they included LYS 198, PHE 336, GLN 191, THR 56, and ILE 52. These findings are consistent with the results from hydrogen bond analysis and the ligand–protein residue interactions derived from clustering analysis.

## 3. Discussion

This study employed molecular docking techniques to screen a series of compounds from the Specs database, aiming to identify potential inhibitors targeting the SLC7A11/xCT axis. Subsequently, the biological activities of these compounds were comprehensively evaluated in HeLa cells. The results revealed that compound 1 effectively disrupted the redox balance within HeLa cells. Further investigations demonstrated that compound 1 targeted the SLC7A11/xCT axis, leading to a significant increase in intracellular ROS levels and ultimately inducing apoptosis in HeLa cells. Moreover, compound 1 exhibited notable inhibitory effects on the migratory capability of HeLa cells at low concentrations, further confirming its potential as an anti-cancer agent. Lastly, validation through three-dimensional tumor spheroid experiments confirmed the cytotoxic effects of compound 1 on HeLa cells and underscored its promising application prospects within the complex tumor microenvironment in vivo.

Molecular dynamics (MD) simulations revealed a stable binding interaction between compound 1 and SLC7A11, impacting the structure and function of SLC7A11. Specifically, RMSD analysis indicated that compound 1 can stably bind to SLC7A11. Upon ligand binding, the average radius of gyration of SLC7A11 remained consistent with its initial conformation, while the radius of gyration in the APO state decreased slightly. This change may be attributed to the contraction of the binding pocket space in the absence of ligand binding. RMSF analysis based on residues showed increased fluctuations in the APO state compared to the ligand-bound systems, possibly due to attractive forces between the ligand and residues near the binding pocket. This was further supported by PCA analysis, which revealed distinct differences in the motion of the binding pocket. Additionally, DCCM analysis showed that the SLC7A11-c1 complex exhibited reduced anti-correlated motions, potentially leading to channel closure and inhibition of cystine transport. Hydrogen bond analysis and clustering results demonstrated that compound 1 formed an extensive hydrogen bonding network with key residues of SLC7A11 and identified major ligand binding conformations. Comparing with the positive control, the binding free energy of compound 1 showed no significant difference, further confirming its stable binding to SLC7A11. It was observed that the residues contributing the most to the binding free energy, such as GLN 191, LYS 198, and PHE 336, exhibited lower RMSF. This observation suggests that favorable interactions between the ligand and the surrounding residues lead to the rigidification of the binding pocket, which in turn promotes a tight association between the ligand and SLC7A11. These findings have provided novel insights into the survival mechanisms of tumor cells and have established a robust theoretical and experimental foundation for the development of therapeutic strategies targeting the SLC7A11/xCT axis.

## 4. Materials and Methods

### 4.1. Chemicals

The compounds in the Specs database were purchased from TOPSCIENCE (Shanghai, China), and erastin (product code: HY-15763) was obtained from MedChemExpress (Shanghai, China). N-acetylcysteine (NAC, product code is N170064) and L-cysteine (product code: C108237) were purchased from Aladdin Reagent Co., Ltd. (Shanghai, China). Calcein-AM (product code: C2013M) was obtained from Beyotime (Beijing, China). All of the above compounds were used as received.

### 4.2. Molecular Docking

Molecular docking analyses were conducted employing the Glide module within the Schrodinger software suite [37,38]. Initially, the crystal structure of SLC7A11 (PDB ID: 7EPZ) [39] underwent preparation and optimization procedures via the Protein Preparation Wizard module. This entailed the addition of hydrogen atoms to amino acids and subsequent optimization and minimization of the protein structure. Subsequently, docking sites were designated based on the positions of native ligands within the receptor, with corresponding docking grids generated accordingly. Small molecules sourced from the Specs database, along with positive control compounds, underwent preparation steps involving hydrogen atom addition and energy optimization utilizing the OPLS3 force field. Following preparation, the compounds were docked onto the protein employing Glide SP (Standard Precision) and Glide XP (Extra Precision) methodologies [37,38,40,41]. The resultant docking analyses involved the assessment of overall binding scores and the characterization of hydrogen bond interactions’ strength between the small molecules and the residues at the receptor’s binding site.

### 4.3. Molecular Dynamics Simulations

Molecular dynamics (MD) simulations were conducted using the Amber 18 software package [36]. The system SLC7A11 was embedded into a pure 1-palmitoyl-2-oleoyl-sn-glycero-3-phosphocholine (POPC) membrane using Maestro. Hydrogen atoms for the proteins were added using the tleap module based on the ff14SB force field [42]. The force field parameters for all candidate compounds were generated using the general AMBER force field (GAFF) [43]. Charges for the small molecules were calculated using Gaussian 16 software [44] with the B3LYP/def2SVP method. The complexes were solvated in a TIP3P water box with a 12.0 Å buffer. Energy minimization of the complexes was performed using 5000 steps of steepest descent followed by 5000 steps of conjugate gradient. The thermalization protocol is delineated into a two-step procedure. In the initial phase, the system is progressively heated to a temperature of 100 K. Subsequently, the system is cautiously warmed to 303 K while maintaining its structural integrity. This was followed by a 5 ns constant-pressure molecular dynamics (MD) simulation conducted to equilibrate the system’s density and dimensions. Finally, 500 ns MD simulations were conducted for all systems under the NVT ensemble at 303 K, with a time step of 2 fs. Temperature was controlled using the Langevin thermostat, and pressure was controlled using the anisotropic Berendsen barostat.

After completing the MD simulations, further analysis of each system’s MD trajectories was performed using the cpptraj program in AMBER, including root mean square deviation (RMSD) analysis, radius of gyration (Rg) analysis, root mean square fluctuation (RMSF) analysis, hydrogen bond analysis, and clustering analysis based on k values. Furthermore, the MD trajectories were subjected to additional analysis using the R package Bio3d 2.4.1.9000 [45], including principal component analysis [46] and dynamic cross-correlation analysis. Visualization of these analyses was performed using software such as PyMOL 2.1.1. Finally, the MM/PBSA module in AMBER was employed to calculate the binding energies of ligands and compounds, and residue-based energy decomposition was performed.

### 4.4. Cell Culture

For this investigation, human cervical cancer cells (HeLa) were sourced from the American Type Culture Collection (ATCC) website. Cell culture essentials such as the cell culture medium, fetal bovine serum (FBS), 25% trypsin-EDTA, and antibiotic solution were procured from Invitrogen (Carlsbad, CA, USA), comprising penicillin and streptomycin. The cells were cultivated under standard conditions at 37 °C with 5% CO_2_ in Dulbecco’s modified Eagle medium (DMEM) supplemented with penicillin/streptomycin, 10% FBS, and glutamine.

### 4.5. Cell Viability Assay

Cells were plated at a density of 2 × 10^3^ cells per well onto a 96-well plate. Following a 24 h incubation period, the culture medium in each well was aspirated and replaced with 100 μL of DMEM containing varying concentrations of compounds in a 96-well microplate. The impact of the compounds on cell viability was evaluated using the CCK-8 assay after incubation at 37 °C for 12, 24, 36, or 48 h.

### 4.6. Cell Scratch Assay

Before cell seeding, two parallel lines were evenly marked on the underside of a 12-well plate using a marker pen. Cells were then seeded into the 12-well plate at a density of 5 × 10^4^ cells per well. The experimental group was treated with varying concentrations (0.5, 1, 2, and 4 μM) of compound 1 or erastin, while the control wells received no treatment (0 μM concentration). Once the cell monolayer reached confluence, a scratch was created by gently dragging a pipette tip across the surface of the cells. Subsequently, the cells were washed three times with PBS and fresh culture medium containing 2% fetal bovine serum was added. The plate was then placed in a 37 °C incubator with 5% CO_2_. Cell migration into the scratch area was observed and recorded under an optical microscope at 20× magnification at 0 and 24 h. This assay was employed to assess the invasiveness of HeLa cells. Each experimental condition was conducted in quadruplicate, and the average migration distance was calculated for analysis.

### 4.7. Migration Assay

HeLa cells (5 × 10^5^ cells) were suspended in DMEM supplemented with 2% FBS at a concentration of 4 μM of compound 1 or erastin and seeded into the upper chambers of Transwell inserts. The lower chambers were filled with the same concentration of the respective compound diluted in medium containing 10% FBS. Control wells received a 0 μM concentration of both substances. Following 24 h of incubation, non-migrated cells in the upper chambers were gently removed using a cotton swab. The cells that had migrated through the membrane and adhered to the lower surface were fixed with 4% paraformaldehyde, stained with crystal violet, and quantitatively analyzed based on the area of cell migration.

### 4.8. ROS Assay

HeLa cells in logarithmic growth phase were collected and seeded at a density of 5 × 10^4^ cells in a 6-well plate. After 24 h, the cells were treated with 25 μM of compound 1 or erastin, and a blank control group was set simultaneously for incubation. The 6-well plate was then placed in a cell incubator at 37 °C with 5% CO_2_ for 24 h. After removing the cell culture medium, the cells were collected in a flow cytometry tube, washed twice with cold PBS, and incubated with a final concentration of 10 µM fluorescent probe DCFH-DA under the condition of 37 °C and 5% CO_2_ for 30 min. The cells were then washed twice with cold PBS, resuspended in 500 µL PBS, and analyzed using a flow cytometer.

### 4.9. Toxicity Assessment of 3D Tumor Spheroids

HeLa cells were suspended in a medium containing 0.75% Matrigel and seeded in 96-well round-bottom ultra-low-attachment plates at a density of 200 cells/well. After incubation for 24 h, HeLa cell spheroids were formed. The HeLa cell spheroids were treated with 5 or 10 μM of compound 1 or 10 μM of erastin in the 96-well round-bottom ultra-low-attachment plates for 6 days. Images were taken every 24 h. All data were analyzed by ImageJ Pro Plus 6.0.

### 4.10. Calcein-AM Staining

HeLa cells (5 × 10^3^ cells/well) were seeded in a 96-well plate and incubated for 24 h. Subsequently, the cells were treated with a certain concentration gradient of compound 1 or co-incubated with 5 mM or 10 mM NAC and compound 1 for 36 h. After that, the cells were washed twice with PBS and then 100 μL of 2 μM calcein-AM solution was added to each well. The fluorescence was observed and images were captured using a fluorescence microscope.

### 4.11. Intracellular GSH Level Assay

HeLa cells were seeded in 5 cm diameter culture dishes with approximately 1 × 10^6^ cells per well using DMEM. After 24 h of cell adhesion, 25 μM of compound 1 or erastin was added to each well, and the cells were further cultured for 24 h. Subsequently, the HeLa cells from each treatment group were collected, and 1 mL of lysis buffer was added to each well to lyse the cells. The samples were centrifuged at 4 °C, 12,000 rpm for 10 min, and the supernatant was collected for GSH determination. The GSH concentration was measured strictly according to the instructions of the GSH detection kit (product code: BC1175) from Beijing Solarbio Science & Technology Co., Ltd. (Beijing, China). The absorbance was read at a wavelength of 412 nm, and the GSH concentration was calculated using a standard curve.

### 4.12. Intracellular Glutamine Level Assay

HeLa cells were seeded in 5 cm diameter culture dishes at approximately 1 × 10^6^ cells per well using DMEM. After 24 h of cell adhesion, 25 μM of compound 1 or erastin was added to each well, and cells were further cultured for another 24 h. Subsequently, HeLa cells from each treatment group were collected, and 1 mL of lysis buffer was added to each well for cell lysis. After lysis, the samples were centrifuged at 12,000 rpm for 10 min at 4 °C, and the supernatant was collected. The samples were analyzed using a glutamine detection kit (product Code: BC5305) from Beijing Solarbio Science & Technology Co., Ltd. A standard curve was plotted to determine the concentration of glutamine in the samples.

## 5. Conclusions

In this study, molecular docking and molecular dynamics simulations were utilized to identify and evaluate compound 1 as a potential inhibitor of the SLC7A11/xCT axis in HeLa cells. The biological activities of compound 1, including the disruption of redox balance, induction of apoptosis, inhibition of cell migration, and cytotoxic effects in three-dimensional tumor spheroid models, underscore its potential as an anti-cancer agent. These findings provide valuable insights into the mechanisms of tumor cell survival and establish a promising foundation for therapeutic strategies targeting the SLC7A11/xCT axis.

## Figures and Tables

**Figure 1 ijms-25-08284-f001:**
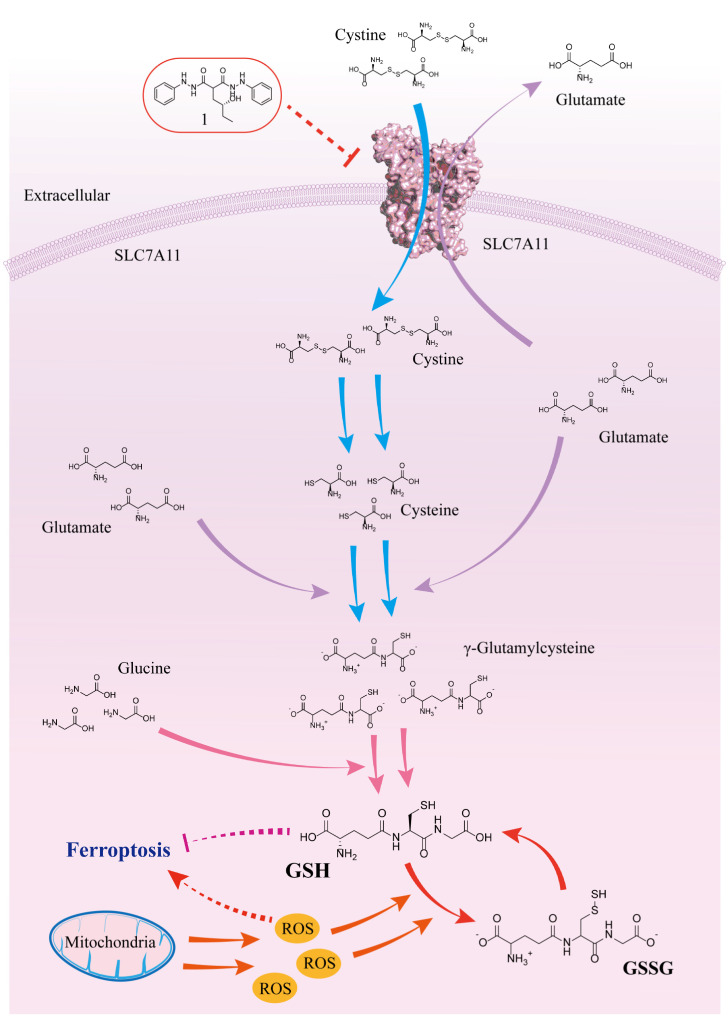
SLC7A11/xCT maintains intracellular GSH/GSSG balance mechanism.

**Figure 2 ijms-25-08284-f002:**
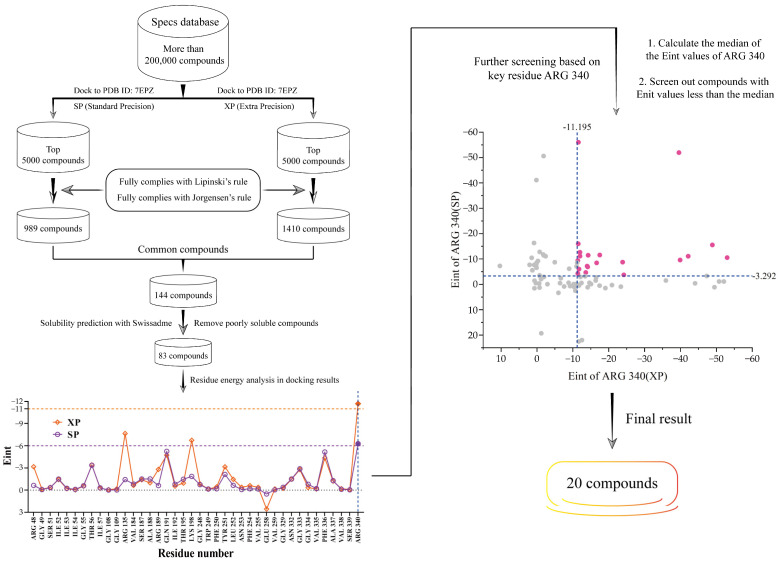
Process of discovering the potential inhibitors targeting the SLC7A11/xCT axis through molecular docking. Eint represents the interaction energy.

**Figure 3 ijms-25-08284-f003:**
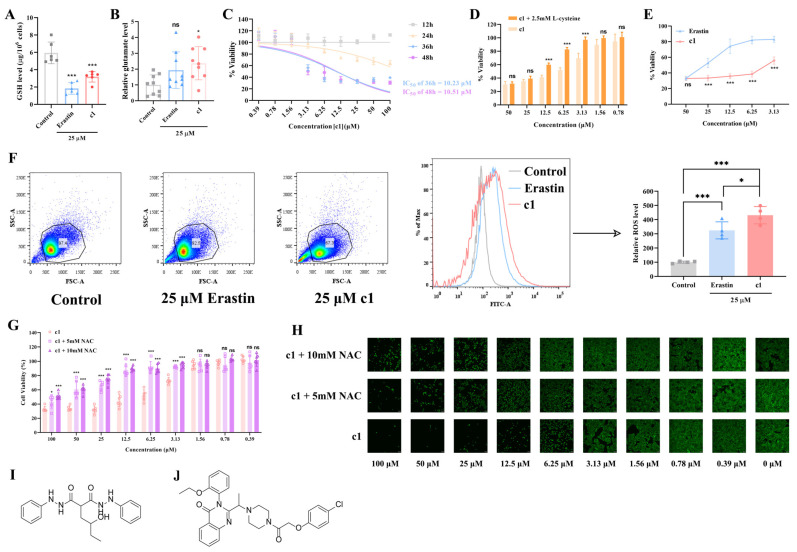
Oxidative/antioxidant balance within the HeLa cells influenced by compound 1. (**A**) After 24 h of drug treatment, the effect of 25 μM compound 1 and erastin on intracellular GSH levels in HeLa cells (n = 6). (**B**) After 24 h of drug treatment, the effect of 25 μM compound 1 and erastin on intracellular relative glutamate levels in HeLa cells (n = 6). (**C**) Cytotoxicity assay results of compound 1 incubated with HeLa cells for 12 h, 24 h, 36 h, and 48 h, respectively (n = 6). (**D**) Cytotoxicity assay results of compound 1 incubated with HeLa cells for 36 h in the presence or absence of 2.5 mM L-cysteine (n = 6). (**E**) Cytotoxicity assay results of compound 1 and erastin at concentrations of 50, 25, 12.5, 6.25, and 3.13 μM, respectively, after 36 h of incubation (n = 5). (**F**) The effect of 25 μM compound 1 and erastin on intracellular ROS levels in HeLa cells (n = 4). (**G**) Cytotoxicity assay results of 5 or 10 mM NAC combined with compound 1 treatment or compound 1 treatment alone after 36 h of incubation (n = 6). (**H**) Calcein-AM staining results of 5 or 10 mM NAC combined with compound 1 treatment or compound 1 treatment alone after 36 h of incubation. (**I**) The 2D structure of compound 1. The magnification is 40×. (**J**) The 2D structure of erastin. The data are presented as mean ± S.D. *, *p* < 0.05; ***, *p* < 0.001, “ns” indicates “non-significant.”

**Figure 4 ijms-25-08284-f004:**
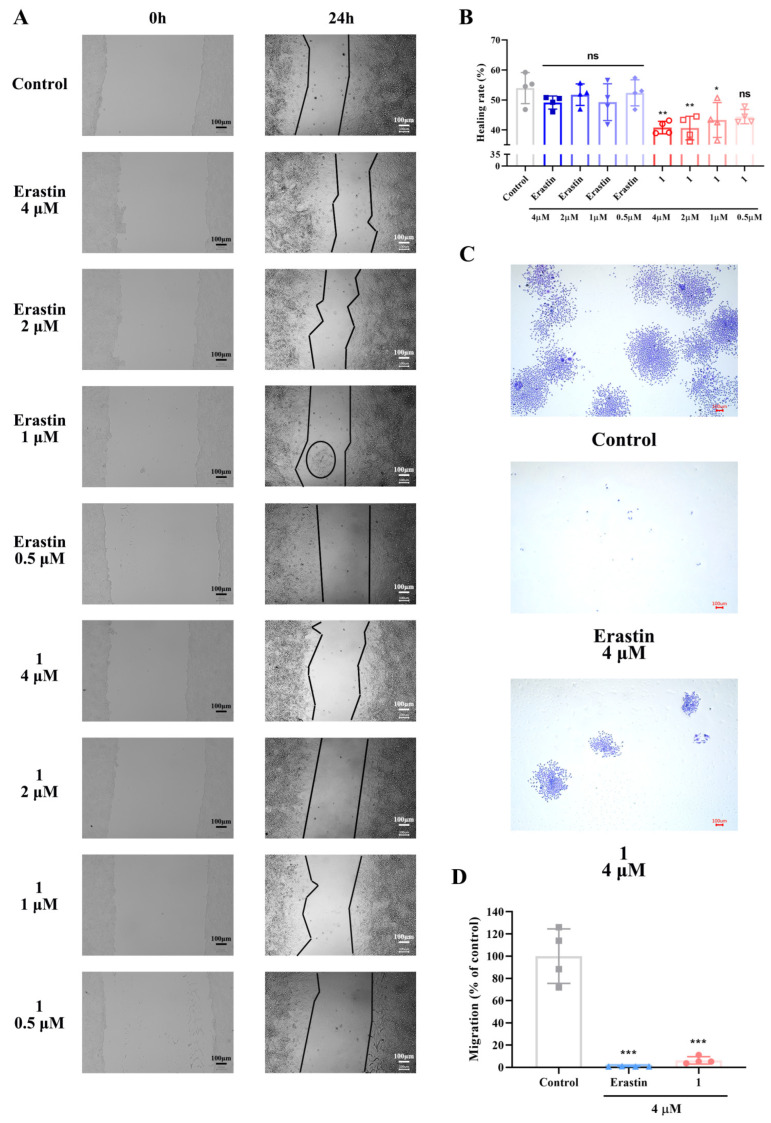
Inhibition of HeLa cell migration by compound 1 at low concentrations. (**A**) Wound healing images. (**B**) Quantitative analysis of scratch wounds treated with different concentrations of compound 1 (0.5, 1, 2, 4 μM) and erastin (0.5, 1, 2, 4 μM) for 24 h (n = 4). (**C**) Transwell migration images. (**D**) Quantitative analysis of Transwell migration after treatment with compound 1 (4 μM) and erastin (4 μM) for 24 h (n = 4). The data are presented as mean ± S.D. *, *p* < 0.05; **, *p* < 0.01, ***, *p* < 0.001, “ns” usually indicates “non-significant.”

**Figure 5 ijms-25-08284-f005:**
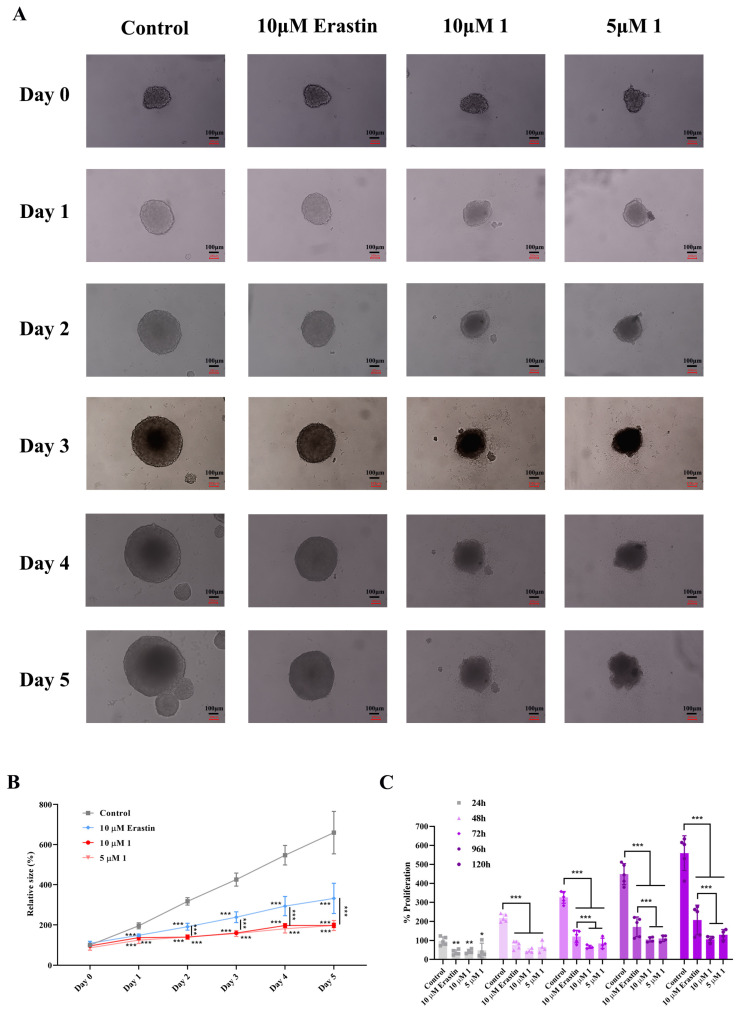
Toxic effects of compound 1 on HeLa three-dimensional spheroids. (**A**) HeLa spheroids were treated with 10 μM erastin or 5 μM and 10 μM compound 1 in a 96-well round-bottom ultra-low-attachment plate. The treatment started on day 0, and images were captured and HeLa spheroid volumes were measured daily from day 0, with six consecutive recordings. (**B**) Changes in HeLa spheroid volumes over 6 days (n = 5). (**C**) Proliferation rates of HeLa spheroid volumes over 6 days (n = 5). The data are presented as mean ± S.D. *, *p* < 0.05; **, *p* < 0.01 and ***, *p* < 0.001.

**Figure 6 ijms-25-08284-f006:**
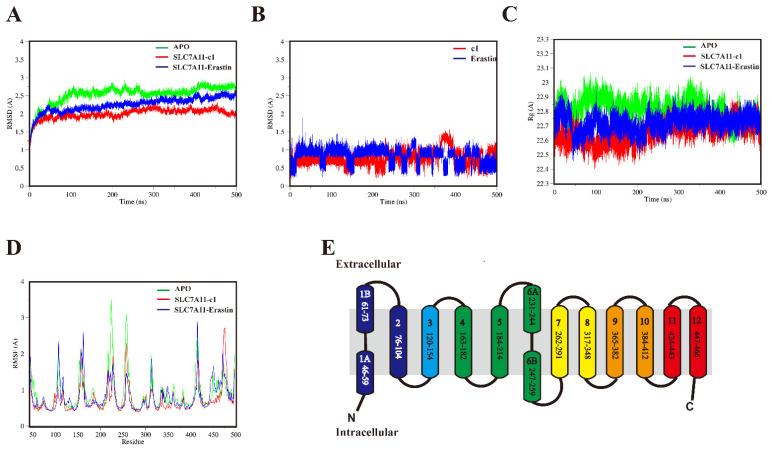
Structural analyses of APO, SLC7A11-c1, and SLC7A11-erastin. (**A**) RMSD of APO, SLC7A11-c1, and SLC7A11-erastin. (**B**) RMSD of compound 1 and erastin. (**C**) Radius of gyration of APO, SLC7A11-c1, and SLC7A11-erastin. (**D**) RMSF of APO, SLC7A11-c1, and SLC7A11-erastin. (**E**) The 2D structure of SLC7A11. APO, SLC7A11-c1, and SLC7A11-erastin (**A**–**D**) are outlined in green, red, and blue lines, respectively.

**Figure 7 ijms-25-08284-f007:**
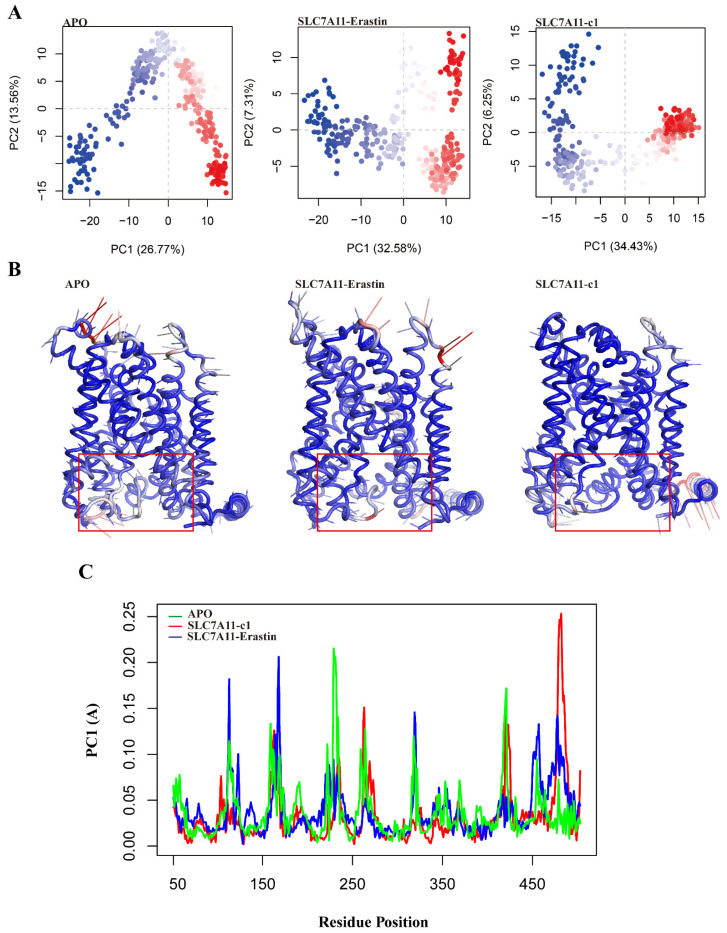
Principal component analysis (PCA) of APO, SLC7A11-c1, and SLC7A11-erastin. (**A**) Essential subspace projection of PC1 vs. PC2 of APO, SLC7A11-c1, and SLC7A11-erastin. The continuous color scale (from blue to white to red) indicates that there are periodic jumps between these conformers throughout the trajectory. (**B**) Porcupine plots represent the motions captured in PC1 of APO, SLC7A11-c1, and SLC7A11-erastin, color scale from blue to red depict low to high atomic displacements. (**C**) Line plots represent the degree of motions captured in PC1 for APO, SLC7A11-c1, and SLC7A11-erastin. The solid box in (**B**) highlights the ligand-binding pocket within the structure.

**Figure 8 ijms-25-08284-f008:**
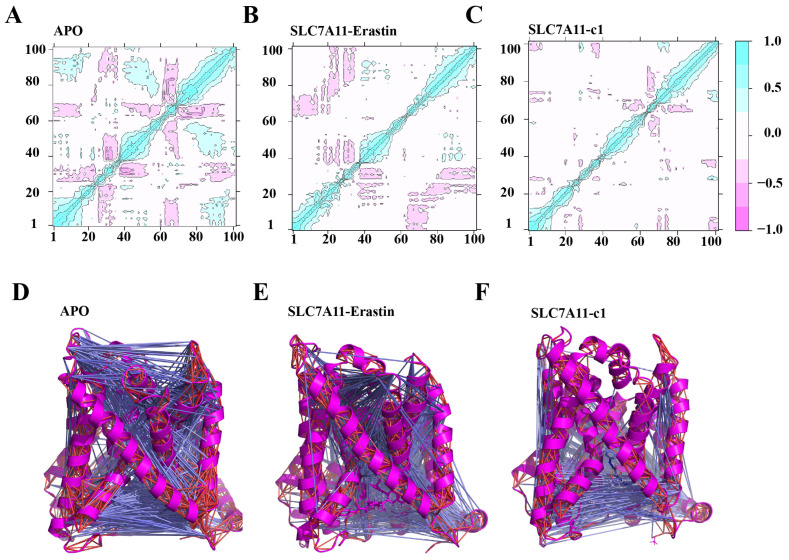
The dynamics cross-correlation matrix (DCCM) plots for APO (**A**,**D**), SLC7A11-c1 (**B**,**E**), and SLC7A11-erastin (**C**,**F**). Red (−1) and blue (+1) correspond to correlated and anti-correlated motions, respectively.

**Figure 9 ijms-25-08284-f009:**
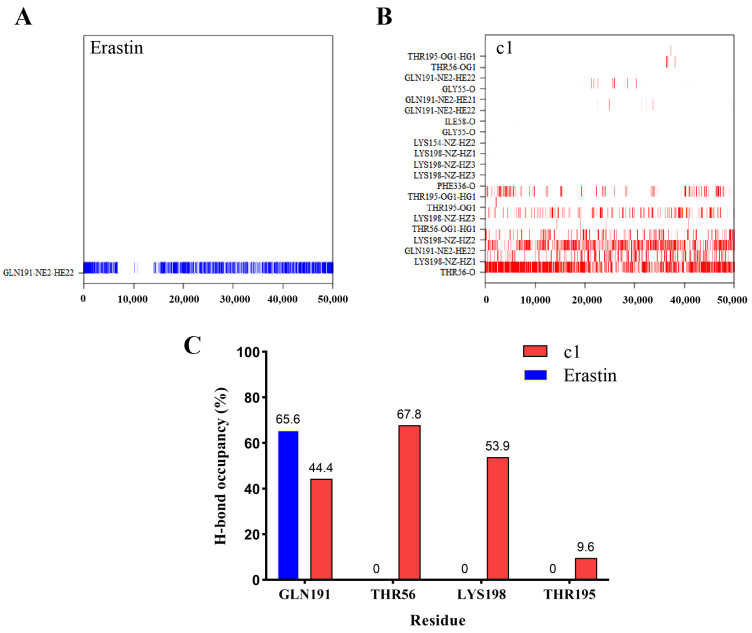
H-bond analysis results. (**A**) Analysis of hydrogen bond interactions between erastin and SLC7A11. (**B**) Analysis of hydrogen bond interactions between compound 1 and SLC7A11. (**C**) H-bond occupancy of each interacting residue in their relevant systems, including SLC7A11-c1, and SLC7A11-erastin.

**Figure 10 ijms-25-08284-f010:**
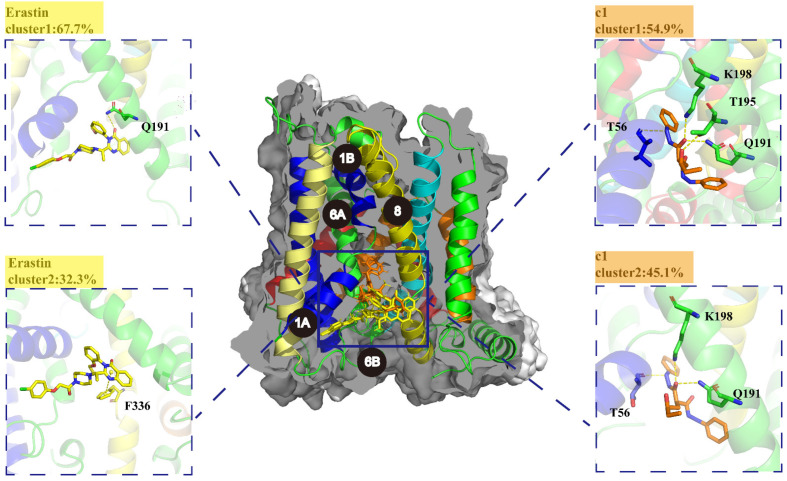
Representative conformation of compound 1(orange) and erastin(yellow) bound to SLC7A11. The key residues of SLC7A11 are in green.

**Figure 11 ijms-25-08284-f011:**
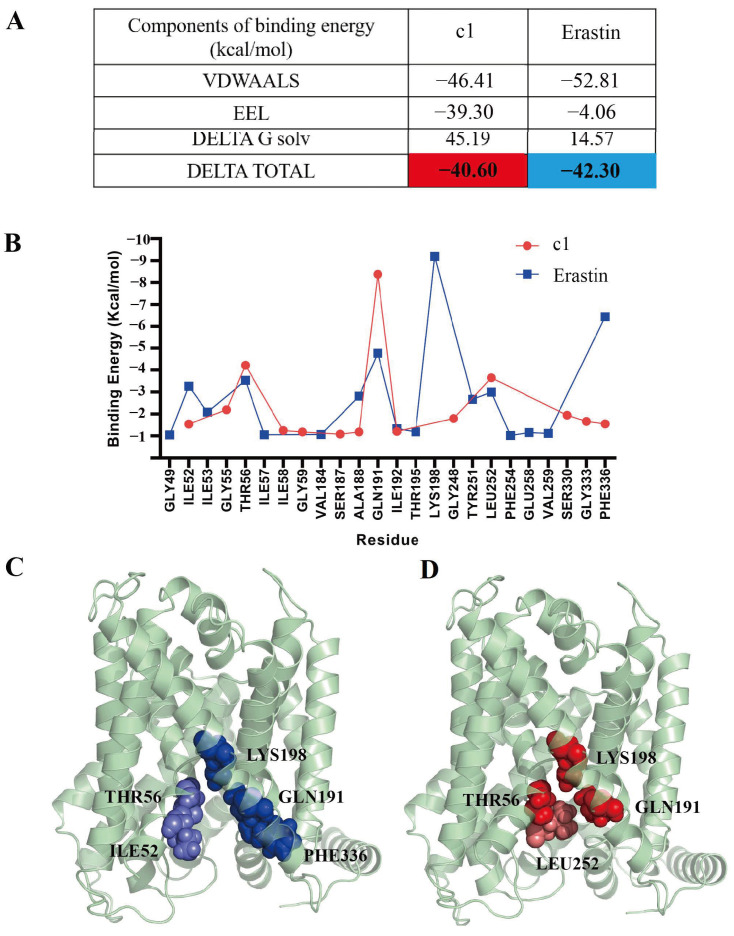
Residue binding energy contribution of SLC7A11-c1 and SLC7A11-erastin. (**A**) Overall binding free energies of compounds with SLC7A11. (**B**) Binding energies of compounds to SLC7A11 residues. (**C**) The positions of residues with high energy contribution bound to erastin. (**D**) The positions of residues with high energy contribution bound to compound 1. The deeper the color, the higher the energy contribution.

## Data Availability

The authors confirm that the data supporting the findings of this study are available within the article and the Appendix A.

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
