# Peer review of "Discovery of the Inhibitor Targeting the SLC7A11/xCT Axis through In Silico and In Vitro Experiments"

_ijms, 2024, doi:10.3390/ijms25158284_

Round 1

Reviewer 1 Report

Comments and Suggestions for Authors

 In this study authors evaluated a compound targeting the SLC7A11/xCT axis named compound 1 and found that, in HeLa cells, compound 1 could reduce intracellular glutathione levels, increase glutamate and ROS levels, and induce cell death. These effects were due to a strong binding affinity with SLC7A11 of compound 1.

The manuscript is interesting and generally well written. Tables are clear but some figures must be improved. My comments are listed below.

Line 54: Figure 1 should be moved early after it is mentioned

Lines 55-56: It deserves to be pointed out that SLC7A11 expression is increased in several tumors and it plays a key role in chemoresistance occurrence (especially to the platinum-derived chemotherapeutics) (see PMID: 38203758). This is an important point to add since it highlights the multifaceted role of this transporter. 

Figure 1: Authors must also show how SLC7A11 inhibits ferroptosis 

Figure 3F and H are unreadable

Figure 8A,B,C: Images are too small

4. Materials and Methods: Authors must provide the product code of all reagents and kits used 

Authors must add the number of replicates (N) in the legend of each figure

Correct superscripts and subscripts throughout the manuscript

Figures must be moved after they are mentioned 

Author Response

Response to Reviewer X Comments

1. Summary

2. Questions for General Evaluation

Reviewer’s Evaluation

Response and Revisions

Does the introduction provide sufficient background and include all relevant references?

Must be improved

[Please give your response if necessary. Or you can also give your corresponding response in the point-by-point response letter. The same as below]

Are all the cited references relevant to the research?

Yes

Is the research design appropriate?

Can be improved

Are the methods adequately described?

Can be improved

Are the results clearly presented?

Must be improved

Are the conclusions supported by the results?

Can be improved

3. Point-by-point response to Comments and Suggestions for Authors

Comments 1: Line 54: Figure 1 should be moved early after it is mentioned

Response 1: Thank you very much for your thorough review of our manuscript and for the valuable suggestions. We have moved Figure 1 to immediately follow the last sentence of the first paragraph in the Introduction section, where we mention the critical role of this mechanism in cancer cell survival.

“This mechanism is crucial for the survival of cancer cells (as illustrated in Figure 1, which depicts the specific mechanism of SLC7A11/xCT in maintaining intracellular glutathione synthesis).”

Comments 2: Lines 55-56: It deserves to be pointed out that SLC7A11 expression is increased in several tumors and it plays a key role in chemoresistance occurrence (especially to the platinum-derived chemotherapeutics) (see PMID: 38203758). This is an important point to add since it highlights the multifaceted role of this transporter.

Response 2:We greatly appreciate the time and effort you have dedicated to reviewing our manuscript and providing valuable feedback.

Upon reviewing the suggested article (PMID: 38203758), we found it to be a comprehensive and insightful study. The article elaborates on the notable upregulation of SLC7A11 in several types of cancers, and its critical function in inhibiting ferroptosis, thereby facilitating cancer cell proliferation, invasion, and chemoresistance. The profundity and clarity presented in this paper have significantly enhanced our understanding and perspective on the subject.

In light of this, we have revised our manuscript to incorporate these important insights. We have added the following statement at the beginning of the second paragraph in the Introduction section: “SLC7A11 is significantly upregulated in various types of cancers18. Research has found that SLC7A11 can inhibit ferroptosis of cancer cells13, 15, 17, 18, including cervical cancer cells16, 19, 20, thereby promoting tumor development.”

References:

13. Koppula, P.;  Zhang, Y.;  Zhuang, L.; Gan, B., Amino acid transporter SLC7A11/xCT at the crossroads of regulating redox homeostasis and nutrient dependency of cancer. Cancer Commun (Lond) 2018, 38 (1), 12.

15. Koppula, P.;  Zhuang, L.; Gan, B., Cystine transporter SLC7A11/xCT in cancer: ferroptosis, nutrient dependency, and cancer therapy. Protein Cell 2021, 12 (8), 599-620.

16. Fang, X.;  Zhang, T.; Chen, Z., Solute Carrier Family 7 Member 11 (SLC7A11) is a Potential Prognostic Biomarker in Uterine Corpus Endometrial Carcinoma. Int J Gen Med 2023, 16, 481-497.

17. Wang, C.;  Liu, H.;  Xu, S.;  Deng, Y.;  Xu, B.;  Yang, T.; Liu, W., Ferroptosis and Neurodegenerative Diseases: Insights into the Regulatory Roles of SLC7A11. Cell Mol Neurobiol 2023, 43 (6), 2627-2642.

18. Fantone, S.;  Piani, F.;  Olivieri, F.;  Rippo, M. R.;  Sirico, A.;  Di Simone, N.;  Marzioni, D.; Tossetta, G., Role of SLC7A11/xCT in Ovarian Cancer. International Journal of Molecular Sciences 2024, 25 (1), 587.

19. Zhou, X.;  Zhao, X.;  Wu, Z.;  Ma, Y.; Li, H., LncRNA FLVCR1-AS1 mediates miR-23a-5p/SLC7A11 axis to promote malignant behavior of cervical cancer cells. Bioengineered 2022, 13 (4), 10454-10466.

20. Yang, J. Y.;  Ke, D.;  Li, Y.;  Shi, J.;  Wan, S. M.;  Wang, A. J.;  Zhao, M. N.; Gao, H., CNIH4 governs cervical cancer progression through reducing ferroptosis. Chem Biol Interact 2023, 384, 110712.

Comments 3: Figure 1: Authors must also show how SLC7A11 inhibits ferroptosis

Response 3: We sincerely appreciate your insightful recommendation regarding the depiction of the mechanism by which SLC7A11 inhibits ferroptosis in Figure 1. This suggestion has significantly enhanced the clarity and accuracy of our manuscript, thereby facilitating a more comprehensive understanding for our readers.

In response to your valuable feedback, we have meticulously revised Figure 1 to incorporate the specific pathways and molecular interactions through which SLC7A11 inhibits ferroptosis. The updated figure now elucidates the role of SLC7A11 in maintaining cystine import, promoting glutathione synthesis, and preventing lipid peroxidation. This modification aims to enhance the mechanistic depiction, providing readers with a more thorough and detailed understanding of the anti-ferroptotic function of SLC7A11.

Comments 4: Figure 3F and H are unreadable

Response 4: We sincerely thank you for your invaluable feedback and meticulous review. Your detailed observations have been instrumental in improving the quality and clarity of our manuscript.

Based on your comments regarding the readability of Figure 3F and 3H, we have taken the following steps to ensure they are clearer and easier to read:

l We have enlarged Figures 3F and 3H.

l We have increased the clarity of these figures to enhance their readability.

We believe these changes will make the figures more accessible to the readers. The revised figures are included in the updated manuscript.

Comments 5: Figure 8A,B,C: Images are too small

Response 5: We sincerely appreciate your timely reminder regarding the size of the images in Figure 8A, B, and C. Your observation is crucial for ensuring that the data is clearly and accurately presented.

In response to your feedback, we have enlarged the images and adjusted the font size in Figure 8A, B, and C to enhance their visibility and readability.

Comments 6: 4. Materials and Methods: Authors must provide the product code of all reagents and kits used

Response 6: We sincerely appreciate your thoughtful feedback. We have revised the relevant sections of the Materials and Methods to include the product codes for all reagents and kits used:

4.1. Chemicals

“The compounds in the Specs database were purchased from TOPSCIENCE (USA), and erastin (Product Code: HY-15763) was obtained from MedChemExpress (USA). NAC (N-acetylcysteine, Product Code: N170064) and L-cysteine (Product Code: C108237) were purchased from Aladdin Reagent Co., Ltd. (Shanghai). Calcein-AM (Product Code: C2013M) was obtained from Beyotime. All of the above compounds were used as received.”

4.11. Intracellular GSH Level Assay

“The GSH concentration was measured strictly according to the instructions of the GSH detection kit (Product Code: BC1175) from Beijing Solarbio Science & Technology Co., Ltd.” 

4.12. Intracellular Glutamine Level Assay

“The samples were analyzed using a glutamine detection kit (Product Code: BC5305) from Beijing Solarbio Science & Technology Co., Ltd.”

Comments 7: Authors must add the number of replicates (N) in the legend of each figure

Response 7: Thank you very much for your insightful feedback. Based on your suggestion, we have added the number of replicates (N) to the respective figure legends in Figures 3, 4, and 5. The specific modifications are as follows:

“Figure 3. Compound 1 (c1) disrupts the oxidative/antioxidant balance within HeLa cells by targeting the SLC7A11/xCT axis. A. After 24 h of drug treatment, the effect of 25 μM compound 1 and erastin on intracellular GSH levels in HeLa cells (n=6). B. After 24 h of drug treatment, the effect of 25 μM compound 1 and erastin on intracellular relative glutamate levels in HeLa cells (n=6). C. Cytotoxicity assay results of compound 1 incubated with HeLa cells for 12 h, 24 h, 36 h, and 48 h, respectively (n=6). D. Cytotoxicity assay results of compound 1 incubated with HeLa cells for 36 h in the presence or absence of 2.5 mM L-cysteine (n=6). E. Cytotoxicity assay results of compound 1 and erastin at concentrations of 50, 25, 12.5, 6.25, and 3.13 μM, respectively, after 36h of incubation (n=5). F. The effect of 25 μM compound 1 and erastin on intracellular ROS levels in HeLa cells (n=4). G. Cytotoxicity assay results of 5 or 10 mM NAC combined with compound 1 treatment or compound 1 treatment alone after 36 h of incubation (n=6). H. Calcein-AM staining results of 5 or 10 mM NAC combined with compound 1 treatment or compound 1 treatment alone after 36 h of incubation. The data are presented as mean ± S.D. *, p < 0.05; **, p < 0.01, ***, p < 0.001.”

“Figure 4. Compound 1 effectively inhibits the migration of HeLa cells at low concentrations. A. Wound healing images. B. Quantitative analysis of scratch wounds treated with different concentrations of compound 1 (0.5, 1, 2, 4 μM) and erastin (0.5, 1, 2, 4 μM) for 24 h (n=4). C. Transwell migration images. D. Quantitative analysis of transwell migration after treatment with compound 1 (4 μM) and erastin (4 μM) for 24 h (n=4). The data are presented as mean ± S.D. *, p < 0.05; **, p < 0.01, ***, p < 0.001.”

“Figure 5. Compound 1 exhibits toxicity towards HeLa three-dimensional spheroids. A. HeLa spheroids were treated with 10 μM erastin or 5 μM and 10 μM compound 1 in a 96-well round-bottom ultra-low attachment plate. The treatment started on day 0, and images were captured and HeLa spheroid volumes were measured daily from day 0, with six consecutive recordings. B. Changes in HeLa spheroid volumes over 6 days (n=5). C. Proliferation rates of HeLa spheroid volumes over 6 days (n=5). The data are presented as mean ± S.D. *, p < 0.05; **, p < 0.01; ***, p < 0.001.”

Comments 8: Correct superscripts and subscripts throughout the manuscript

Response 8: Thank you very much for your valuable feedback, we have thoroughly reviewed and corrected all superscripts and subscripts in the manuscript. The specific corrections we made are as follows:

l Corrected “CO2” to “COâ‚‚” throughout the manuscript.

l Corrected “IC50” to “ICâ‚…â‚€.”

l Standardized cell counts, for example, changing “2×103 cells” to “2×10³ cells.”

l Ensured that all other instances of superscripts and subscripts are properly formatted in the text, figures, and tables.

Thank you once again for your constructive suggestion.

Comments 9: Figures must be moved after they are mentioned

Response 9: We appreciate your insightful feedback regarding the clarity and organization of our manuscript, we conducted a thorough review of the manuscript to ensure that the discussion of each result is closely aligned with its respective figure. We specifically addressed instances where the figures and their explanatory text were not in close proximity, and made necessary adjustments to improve the flow and readability of the manuscript. The specific modifications we made are as follows:

l Figure 1: We have repositioned Figure 1 to immediately follow the last sentence of the first paragraph in the Introduction section.

l Figure 4: To facilitate a more seamless reading experience, Figure 4 has been moved to directly follow the section 2.3, titled "Compound 1 effectively inhibits the migration of HeLa cells at low concentrations."

l Additionally, we carefully reviewed and adjusted the placement of other figures and their corresponding text throughout the manuscript to ensure consistency and improve the overall readability.

4. Response to Comments on the Quality of English Language

5. Additional clarifications

Reviewer 2 Report

Comments and Suggestions for Authors

This manuscript encloses the in silico discovery of new SLC7A11/xCT axis inhibitor by structure-based high-throughput virtual screening. The biological activity of hit compound was evaluated in vitro on cervical cancer cells. The promising results were obtained since compound 1 was able to reduce the gluthatione levels and thereby disrupt the redox balance leading to oxidative stress and ultimately cell death. Cells migration was also significantly interrupted by compound 1. A known SLC7A11 inhibitor Erastin was used as standard compound and in most experiments comparable results were obtained, however, in some instances compound 1 showed better results than erastin.  Additionally, molecular dynamics was used to computationally predict the stability of ligand-protein complexes and most contributing interactions between compound 1 and residues in the binding pocket of the SLC7A11.

In general, this manuscript is well written and methodology is well conceived. There are however few major changes that needs to be corrected prior the publication.

1.       Figure captions in some instances should be corrected. For example: Figure 2. Potential inhibitors targeting the SLC7A11/xCT axis discovered through molecular docking. I recommend changing it to: process of discovering the potential inhibitors targeting the SLC7A11/xCT axis through molecular docking or similar, since it depicts the process not the actual compounds. Figure 3. Figure 3. Compound 1 (c1) disrupts the oxidative/antioxidant balance within HeLa cells by targeting the SLC7A11/xCT axis. I suggest changing it to: Oxidative/antioxidant balance within the HeLa cells influenced by the compound 1…It should not start with the result or subject of the experiment. The rest of the captions should be checked and corrected accordingly.

2.       The same is for the subtitles. For example: 2.1. Potential inhibitors targeting the SLC7A11/xCT axis discovered through molecular docking it should be corrected to: “The discovery of potential SLC7A11/xCT axis inhibitors…  or “2.2. Compound 1 disrupts the oxidative/antioxidant balance within HeLa cells by targeting the SLC7A11/xCT axis” should be something like: In vitro evaluation of the effect compound 1 has on the oxidative/antioxidant balance within HeLa cells or similar.  The object of the specific experiment should be in the subtitle not the result. The rest of the subtitles should be checked and corrected accordingly. As I can see, in the computational part of the manuscript there are no such errors.

3.       Discussion part, lines 422-423. The authors stated: Further investigations demonstrated that 422 compound 1 selectively targeted the SLC7A11/xCT axis… selectively compared to what? If there are no data to corroborate this statement it should be removed.

4.       Supplementary file: Please provide better copies of NMR, IR and LC/MS spectra, since most of these are barely visible. Also Figure captions, it cannot be for every different spectra figure caption: The molecular composition of compound… The correct figure caption for each different spectra would be: 1H NMR spectra of compound … or IR spectra of compound …

5.       I would suggest that the image of compound 1 and erastin should be added in the main part of the manuscript.

6.       The Conclusion part is missing!

Comments on the Quality of English Language

1.       In the introduction part lines 77-79, sentence “This study aims to discover targeted inhibitors of the SLC7A11/xCT axis through 77 structure-based high-throughput virtual screening, disrupting the redox balance in cervical cancer cells and inducing cell death.” looks like something that would ChaTGPT wrote. It should be corrected for example: “ This study aims to discover targeted inhibitors of the SLC7A11/xCT axis through 77 structure-based high-throughput virtual screening, that would potentially be able to disrupt the redox balance in cervical cancer cells and inducing cell death.”

2.       Line 77, the paragraph starts with the “this study aims” and finishes with “this study aimed” with the future tense in between, line 84, the “will be used” should be corrected to “is used” or “was used”. Overall the grammar check would be useful here.

Author Response

Response to Reviewer X Comments

1. Summary

2. Questions for General Evaluation

Reviewer’s Evaluation

Response and Revisions

Does the introduction provide sufficient background and include all relevant references?

Yes

[Please give your response if necessary. Or you can also give your corresponding response in the point-by-point response letter. The same as below]

Are all the cited references relevant to the research?

Yes

Is the research design appropriate?

Yes

Are the methods adequately described?

Yes

Are the results clearly presented?

Can be improved

Are the conclusions supported by the results?

Can be improved

3. Point-by-point response to Comments and Suggestions for Authors

Comments 1: Figure captions in some instances should be corrected. For example: Figure 2. Potential inhibitors targeting the SLC7A11/xCT axis discovered through molecular docking. I recommend changing it to: process of discovering the potential inhibitors targeting the SLC7A11/xCT axis through molecular docking or similar, since it depicts the process not the actual compounds. Figure 3. Figure 3. Compound 1 (c1) disrupts the oxidative/antioxidant balance within HeLa cells by targeting the SLC7A11/xCT axis. I suggest changing it to: Oxidative/antioxidant balance within the HeLa cells influenced by the compound 1…It should not start with the result or subject of the experiment. The rest of the captions should be checked and corrected accordingly.

Response 1: We sincerely thank you for your invaluable feedback and meticulous review. Based on your suggestions regarding the figure captions in our manuscript, we have reviewed and made the following modifications:

Figure 2 caption has been changed to: “Figure 2. Process of discovering the potential inhibitors targeting the SLC7A11/xCT axis through molecular docking.”

Figure 3 caption has been changed to: “Figure 3. Oxidative/antioxidant balance within the HeLa cells influenced by the compound 1.”

Figure 4 caption has been changed to: “Figure 4. Inhibition of HeLa cell migration by compound 1 at low concentrations.”

Figure 5 caption has been changed to: “Figure 5. Toxic effects of compound 1 on HeLa three-dimensional spheroids.”

We appreciate your thorough review and constructive comments, which have greatly contributed to improving the quality of our manuscript. Thank you once again for your valuable input.

Comments 2: The same is for the subtitles. For example: 2.1. Potential inhibitors targeting the SLC7A11/xCT axis discovered through molecular docking it should be corrected to: “The discovery of potential SLC7A11/xCT axis inhibitors…  or “2.2. Compound 1 disrupts the oxidative/antioxidant balance within HeLa cells by targeting the SLC7A11/xCT axis” should be something like: In vitro evaluation of the effect compound 1 has on the oxidative/antioxidant balance within HeLa cells or similar.  The object of the specific experiment should be in the subtitle not the result. The rest of the subtitles should be checked and corrected accordingly. As I can see, in the computational part of the manuscript there are no such errors.

Response 2: We sincerely thank you for your invaluable feedback and meticulous review. Your detailed suggestions have been instrumental in improving the quality and clarity of our manuscript.

Based on your insightful comments regarding the subtitles, we have made the following specific modifications:

l 2.1. The discovery of potential SLC7A11/xCT axis inhibitors.

l 2.2. In vitro evaluation of the effect compound 1 has on the oxidative/antioxidant balance within HeLa cells.

l 2.3. In vitro evaluation of the inhibitory effects of compound 1 on HeLa cell migration at low concentrations.

l 2.4. Assessment of the toxic effects of compound 1 on HeLa three-dimensional spheroids.

Comments 3: Discussion part, lines 422-423. The authors stated: Further investigations demonstrated that 422 compound 1 selectively targeted the SLC7A11/xCT axis… selectively compared to what? If there are no data to corroborate this statement it should be removed.

Response 3: We sincerely thank you for your invaluable feedback and meticulous review. Your detailed suggestions have been instrumental in improving the quality and clarity of our manuscript.

Based on your insightful comments regarding the discussion part, lines 422-423, we have removed the term “selectively” to ensure accuracy and clarity. The revised sentence now reads:

“Further investigations demonstrated that compound 1 targeted the SLC7A11/xCT axis, leading to a significant increase in intracellular ROS levels and ultimately inducing apoptosis in HeLa cells.”

Comments 4: Supplementary file: Please provide better copies of NMR, IR and LC/MS spectra, since most of these are barely visible. Also Figure captions, it cannot be for every different spectra figure caption: The molecular composition of compound… The correct figure caption for each different spectra would be: 1H NMR spectra of compound … or IR spectra of compound …

Response 4: We sincerely thank you for your invaluable feedback and meticulous review. Your detailed suggestions have significantly contributed to the improvement of our manuscript and supplementary materials. However, we would like to provide further context regarding this matter:

l Source of Compounds: The compounds used in this study (compounds 1-19) were all purchased from Specs via Shanghai Tasu Company. The 1H NMR spectra included in our supplementary materials were originally supplied by Specs.

l Supplier Communication: We have previously communicated with Specs regarding the clarity of the provided spectra, and unfortunately, we have not received a satisfactory response addressing these concerns.

l Laboratory Limitations: Due to the constraints of our current laboratory facilities, we lack the advanced instrumentation required to independently reproduce or enhance the clarity of the 1H NMR spectra. Specifically, our laboratory does not possess high-resolution NMR spectrometers or related analytical equipment.

l Assurance of Data Integrity: Despite these limitations, we assure you that the experiments were conducted using the compounds as received from the supplier without any modifications. We stand by the reliability and authenticity of all experimental and computational results presented in this study. We are committed to transparency and scientific rigor in our research and are willing to provide any further information or clarifications as needed.

Based on your suggestion, we have modified the figure captions in the supplementary files to accurately reflect the specific types of spectra. The captions for Figures S5-S23 have been updated to “1H NMR spectra of compound…”. This change ensures that each figure is clearly identifiable and correctly categorized.

We deeply appreciate your understanding regarding the challenges we face with the clarity of certain spectra and your ongoing support of our efforts. We hope that our detailed explanation provides adequate context and assurance of the integrity of our work. While our current laboratory conditions limit our ability to provide higher clarity spectra, we remain dedicated to the pursuit of scientific excellence and will address this issue further as our laboratory capabilities improve in the future.

Thank you once again for your thorough review and valuable suggestions. We look forward to any further feedback you may have.

Comments 5: I would suggest that the image of compound 1 and erastin should be added in the main part of the manuscript.

Response 5: We sincerely thank you for your valuable suggestion. The specific changes made are as follows:

l Added the 2D structures of compound 1 (Figure 3I) and erastin (Figure 3J) to Figure 3.

l Updated the caption of Figure 3 to include: “I. The 2D structure of compound 1. J. The 2D structure of erastin.”

l Added a discussion in the first paragraph of section 2.2: “Meanwhile, the 2D structures of compound 1 and erastin are also shown in Figures 3I and 3J, respectively.”

Comments 6: The Conclusion part is missing!

Response 6: We sincerely thank you for your timely reminder. This section can now be found immediately following section 4.

“5. Conclusion

In this study, molecular docking and molecular dynamics simulations were utilized to identify and evaluate compound 1 as a potential inhibitor of the SLC7A11/xCT axis in HeLa cells. The biological activities of compound 1, including the disruption of redox balance, induction of apoptosis, inhibition of cell migration, and cytotoxic effects in three-dimensional tumor spheroid models, underscore its potential as an anticancer agent. These findings provide valuable insights into the mechanisms of tumor cell survival and establish a promising foundation for therapeutic strategies targeting the SLC7A11/xCT axis.”

4. Response to Comments on the Quality of English Language

Point 1: In the introduction part lines 77-79, sentence “This study aims to discover targeted inhibitors of the SLC7A11/xCT axis through 77 structure-based high-throughput virtual screening, disrupting the redox balance in cervical cancer cells and inducing cell death.” looks like something that would ChaTGPT wrote. It should be corrected for example: “ This study aims to discover targeted inhibitors of the SLC7A11/xCT axis through 77 structure-based high-throughput virtual screening, that would potentially be able to disrupt the redox balance in cervical cancer cells and inducing cell death.”

Response 1: We deeply appreciate your meticulous guidance regarding the language used in our manuscript. Following your advice, we have revised the first sentence in the last paragraph of section 1.

“This study aims to discover targeted inhibitors of the SLC7A11/xCT axis through structure-based high-throughput virtual screening, that would potentially be able to disrupt the redox balance in cervical cancer cells and induce cell death.”

Point 2: Line 77, the paragraph starts with the “this study aims” and finishes with “this study aimed” with the future tense in between, line 84, the “will be used” should be corrected to “is used” or “was used”. Overall the grammar check would be useful here.

Response 2: We appreciate your keen observations and suggestions. Following your advice, we have revised the first sentence in the last paragraph of section 1. 

“This study aims to discover targeted inhibitors of the SLC7A11/xCT axis through structure-based high-throughput virtual screening, that would potentially be able to disrupt the redox balance in cervical cancer cells and induce cell death.”

“Furthermore, molecular dynamics simulation was used to analyze and compare the affinity of compounds with the positive control erastin for SLC7A11.”

Additionally, we conducted a thorough grammar check on the entire document to rectify any other inconsistencies and ensure overall grammatical accuracy. The revised text should now consistently reflect the appropriate tense and maintain the clarity of our findings.

5. Additional clarifications

Round 2

Reviewer 1 Report

Comments and Suggestions for Authors

the manuscript has been significantly improved and can be accepted in the present form